# Development of a Web App to Convert Blood Insulin Concentrations among Various Immunoassays Used in Horses

**DOI:** 10.3390/ani13172704

**Published:** 2023-08-24

**Authors:** Julien Delarocque, Karsten Feige, Harry B. Carslake, Andy E. Durham, Kerstin Fey, Tobias Warnken

**Affiliations:** 1Clinic for Horses, University of Veterinary Medicine Hannover, Foundation, Bünteweg 9, 30559 Hannover, Germany; 2Institute of Infection, Veterinary and Ecological Sciences, University of Liverpool, Leahurst, Neston CH64 7TE, Cheshire, UK; 3The Liphook Equine Hospital, Forest Mere, Liphook GU30 7JG, Hampshire, UK; 4Equine Clinic, Internal Medicine, Faculty of Veterinary Medicine, Justus-Liebig-University of Giessen, Frankfurter Str. 126, 35392 Giessen, Germany

**Keywords:** insulin dysregulation, hyperinsulinaemia, equine metabolic syndrome, laminitis, cut-off, immunoassay

## Abstract

**Simple Summary:**

In horses, the hormone insulin is frequently measured in blood samples, as increased concentrations can lead to a painful condition of the hoof called laminitis. The early detection of an increased insulin concentration is essential for animal welfare. There is considerable disagreement between different measurement methods (assays) for insulin concentration, however, meaning that the threshold for diagnosis must be determined for each assay individually. To alleviate this requirement, we derived formulas from previous assay comparisons, to convert the values from one assay to another, and made them available through a free web app. Over a wide range of commonly used insulin assays, veterinarians can now compare their measurements to previously published values. Scientists can also use the app to compare publications using different assays.

**Abstract:**

The measurement of the blood insulin concentration, and comparison to cut-offs, is essential in diagnosing insulin dysregulation, a common equine endocrinopathy. However, different insulin assays provide disparate results. We aimed to ease comparison between assays by compiling original and published data into a web app to convert insulin measurements from one assay to another. Data were available for ADVIA Centaur insulin chemiluminescent immunoassay (CLIA), Beckman Coulter insulin radioimmunoassay (RIA), Immulite 1000 CLIA, Immulite 2000 CLIA, Immulite 2000 XPi CLIA, Mercodia equine insulin enzyme-linked immunosorbent assay (ELISA), and Millipore porcine insulin RIA. Linear models were fitted for 13 assay pairs using non-decreasing splines, and integrated into this app. Assay comparisons including data from several studies showed a lower performance. This indicates technical variation between laboratories, which has not been described before, but is relevant when diagnostic measurements and cut-offs are provided by different laboratories. Nevertheless, the models’ overall high performance (median *r*^2^ = 0.94; range 0.57–1.00) supports their use to interpret results from diagnostic insulin measurements when the reference assay is unavailable, and to compare values obtained from different assays.

## 1. Introduction

In horses, elevated insulin levels are frequently associated with insulin dysregulation (ID) [1]. Affected horses display postprandial, and sometimes basal, hyperinsulinaemia, which puts them at an increased risk of developing laminitis [2,3]. This condition is painful, can warrant euthanasia [4], and is attributable to ID in about 90% of cases seen in equine hospitals [5]. Therefore, recognizing horses at risk, to implement adequate management and avoid laminitis, is important to animal welfare.

Many protocols have been described to diagnose ID [6]. They generally rely on the oral or intravenous administration of a standardized glucose meal or bolus, respectively [7]. Insulin-dysregulated animals will subsequently present aberrant degrees of hyperinsulinaemia, and are often identified using cut-offs for blood insulin concentrations at predetermined time points [8,9,10,11,12,13]. However, due to the sometimes considerable discrepancies in the results provided by different insulin assays, the cut-offs and the general diagnostic value of the measured blood insulin concentrations must be considered assay-specific [14,15,16,17,18,19,20].

The chemiluminescent immunoassay (CLIA), radioimmunoassay (RIA), and enzyme-linked immunosorbent assay (ELISA) are mainly used to measure insulin in horses. Despite differences in amino acid sequences in different species [21], most of these assays were developed for the measurement of human insulin. Indeed, the WHO currently offers no standard for equine insulin as it does for human, bovine, or porcine insulin [22], which may impair the development of assays relying on anti-equine-insulin antibodies. Moreover, the standardization of insulin assays, as initiated by several bodies, including the American Diabetes Association, is still in progress in humans [23,24,25], and remains elusive in horses. While it may not be required for established diagnostic tests, the standardization of equine insulin assays would probably greatly facilitate the research into equine endocrinology.

Although they offer insulin measurements to the equine practitioner, laboratories rarely mention which assay they use, nor that the results are only comparable to cut-offs established for that specific assay. Furthermore, contrarily to a common assumption, diagnostic thresholds may be different even within the same assay family [26].

Several researchers have compared and assessed the insulin immunoassays used in scientific projects. The present manuscript aimed to harness these data to provide an accessible way of comparing the insulin concentrations from various assays, thus enabling the comparison of publications using different assays, and the use of cut-offs established for other assays.

## 2. Materials and Methods

### 2.1. Assay Comparison Data

The data were obtained from the authors’ archives, and selected publications providing assay comparisons as tables or plots. In the latter case, the data were extracted using custom scripts and/or the publicly available tool: https://plotdigitizer.com/app (accessed on 14 December 2022). The extracted data were checked for plausibility through the comparison of the obtained values with the original plots.

The publications were selected arbitrarily, depending on the estimated effort to extract the data from the plots, and their subjective value—the more valuable datasets being those obtained with commonly-used assays, or connected to assays already available through other datasets.

The data provided by the authors were obtained for diagnostic purposes, or from experiments approved by the authors’ respective home university and/or state office for animal protection upon the obtention of informed consent from the owners. The source of each dataset is detailed in the Results section. Only samples without exogenous insulin were retained. Samples outside of the quantification limits were removed, but diluted samples in those limits were kept. All sources provided insulin concentrations as µIU/mL; however, conversion factors between mass-based and bioefficacy-based units may have been applied beforehand.

### 2.2. Estimation of Conversion Formulae

To allow for non-linear relationships, non-negative least-squares models with a monotonically non-decreasing spline predictor (I-spline) were fitted for each assay pair (with distinct models for assay A to B, and B to A) using R version 4.3.0 [27], and the ‘nnls’ [28] and ‘splines2’ [29] R-packages. This model form ensures that the relationship between two assays is represented as a monotonically increasing function, which is not guaranteed by other modelling approaches. All models were inspected visually, to ensure an adequate fit and the absence of influential outliers. All samples were considered independent, although, in some cases, several samples per individual were present in the data.

### 2.3. Integration into a Web App

The model formulae and prediction interval calculation formulae were extracted from the final models to be included in the web app. There, the predicted values and prediction intervals were calculated for every available assay via the JavaScript [30] evaluation of the corresponding mathematical formulae, after the injection of the insulin value to convert. The upper prediction limit is set to the highest insulin value from the original dataset, minus one (to avoid artefacts from the splines), preventing any extrapolation of the conversion formula above the range of available data.

Apart from the conversion module, the app is written in plain HTML and CSS [31,32]. It contains background information on its purpose, limitations, and underlying methodology, and is accessible at www.equine-insulin-converter.org, accessed on 2 July 2023. The source code is available at github.com/jkdel/insulin-converter, accessed on 2 July 2023, under an Apache 2.0 license [33].

## 3. Results

### 3.1. Available Assays and Data Sources

Data were available for four CLIAs, three RIAs, and one ELISA (Table 1). Eleven sources were used—some comparing several pairs of assays—resulting in 13 assay comparisons (Table 2).

Seven of the data sources are publications from the last twelve years. Data were either extracted from the plots, or already available to the authors. The authors provide four previously unpublished sources. The non-diagnostic samples of the additional sources were obtained during experiments approved by the ethics committee and/or competent state office for animal protection; the respective file numbers are provided alongside each source (Table 2).

### 3.2. Assay Pairs

The pairs of assays compared in the data sources are presented in Figure 1. Several sources were combined into single comparisons (e.g., the Mercodia Equine vs. Immulite 1000 comparison combines four different sources, as shown in Table 2).

Figure 2 presents the data and derived polynomial models. There are thirteen assay pairs. A rule according to which assays of the same family display a smaller absolute and relative deviation (i.e., closer to the identity line) is not recognizable. For example, the three RIAs provide higher (Millipore Porcine), lower (Beckman Coulter), or similar (Coat-a-Count) values than a common reference (Mercodia Equine).

### 3.3. Web App Usage

The web app, available at www.equine-insulin-converter.org, accessed on 2 July 2023, contains a short introduction to its scope, and a conversion tool, and ends with background information on the methodology. The conversion tool aims to be self-explanatory. Figure 3 illustrates how the user can input an insulin concentration for any of the available assays. The range of possible values is initially shown in grey on the right of the input box. Once the “Enter” key is pressed on the keyboard, or the user clicks on the “Convert!” button, the app performs the required calculations. The estimated corresponding insulin concentrations and 95% prediction intervals are provided for assays with enough data in the algorithm, as illustrated in Figure 1. Prediction intervals are wider than classical confidence intervals, because they describe the uncertainty around new predictions, and not the available data.

### 3.4. Example: Comparison of Published Cut-Offs

A list of published cut-offs for basal insulin, presented in Table 3, was converted to values of the Mercodia equine insulin assay (Mercodia AB, Sylveniusgatan 8A, SE-754 50 Uppsala, Sweden), used as a common reference because of the large number of comparable assays. This enables the comparison of these cut-offs, although some sources of variation remain. For the predictor ‘basal insulin’, the outcomes were insulin resistance and/or dysregulation or laminitis. As there may be some overlap between the thereby defined groups, a statistical method is required, to determine the optimal cut-off. For instance, Youden’s index can be used to identify the point maximizing the sum of the cut-off specificity and sensitivity.

## 4. Discussion

An easy-to-use, open-source web app combining insulin assay comparison data was developed. The app enables the comparison of insulin measurements from different assays in a wide range of concentrations.

### 4.1. Use and Limitations of the App

Comparisons of insulin values from different assays can be required when assessing the agreement between scientific works, or interpreting diagnostic tests. While it is preferable to compare values obtained via the same assay, many assays are used around the world, and the present app alleviates this requirement, at the cost of increased uncertainty around the measurements. As a result, the converted values must be considered with care, and only in conjunction with clinical signs of ID.

Some of the publications used as data sources provide sufficient data for manual conversion between one assay and another. However, as well as automating the calculations, the present app provides a more complete range of assays than any individual publication, and yields prediction intervals indicating the uncertainty around the converted estimates.

The combination of several data sources in the comparison of two assays resulted in increased sample sizes, as well as an increased heterogeneity. While this leads to broader prediction intervals, due to added technical variation, it may improve the conversion formula’s robustness. Consequently, the chances of the conversion tool performing well when used with values provided by a random laboratory should be increased (in contrast to performing well only for the laboratory providing the data used to create the app). It should also be noted that, even with the same assay, inter-assay and inter-laboratory variation must be taken into account when using cut-offs established in a different laboratory. In such cases, parts of the uncertainty due to the conversion may be compensated for, through the provision of a conversion formula closer to the true mean relationship between the assays.

Individual factors affecting some insulin immunoassays more than others cannot be excluded. Although the mathematical models in the app do not account for repeated measures within individuals, these data were often neither presented nor accounted for in the original publications. Nonetheless, the inter-individual correlations showed a negligible effect on the conversion formulae in prior tests on the additional datasets provided by the authors. Samples containing exogenous insulin were excluded in the present study because insulin assay performances are species-specific, and exogenous insulin typically differs from equine insulin in its amino acid sequence, and may also differ in its immunoreactivity. Conversely, the app will not provide adequate results for future samples containing exogenous insulin. Finally, the included assays had different analytical ranges, which may require dilutions. As the range in which dilution is required should remain the same for a given assay, and non-linear relationships were permitted in our mathematical approach, we did not control for the effect of dilution, although the recovery on dilution varies among assays.

As conversion is possible up to around 200 µIU/mL on the Mercodia Equine scale for most assays, the usable range of insulin concentrations should suffice to distinguish insulin dysregulated from normo-insulinemic animals, even if dynamic tests are used. Nevertheless, the range of usable blood insulin concentrations could be increased in future versions of the app, through the addition of more data.

Despite the usefulness of the app, cut-offs are most reliable for the assay they were established for. When that assay is not available, we recommend converting the cut-off value, rather than the result, in the diagnostic sample, because the sources of variation are less controlled in the field than in the experimental setting used to determine the cut-off. It should be noted that the conversion formulas are not reciprocal, because they are estimated separately for each direction of conversion (A to B and B to A). Therefore, back-conversions will not be exactly equivalent to the original values. Similarly, chained conversions (when a direct comparison between two assays is not available in the app, and the user relies on a third assay as an intermediate (A to B to C)) will result in increased uncertainty, beyond the provided prediction intervals.

### 4.2. Comparison of Cut-Off Values

As shown above with the basal insulin cut-offs for the diagnosis of ID or a propensity towards laminitis (Table 3), studies providing such cut-offs have many sources of variability, apart from the insulin assay used.

Firstly, the outcome used to establish the cut-off must be considered. Some studies relied on diagnostic tests, such as the euglycemic hyperinsulinaemic clamp (EHC) [35] or combined glucose–insulin test (CGIT) [36], to detect insulin-resistant horses, while others were conducted prospectively, with laminitis as an outcome [37,38,39]. Another study reported a confidence interval for basal insulin in healthy ponies [41]. It is interesting that the intravenous tests provided lower cut-offs than those obtained from a healthy population, or when subsequent laminitis was considered. This may corroborate the higher sensitivity of dynamic testing protocols, but also warrants further investigation into the likeness of dynamic testing protocols with naturally occurring insulinaemic stimuli [42]. Additionally, the higher cut-offs associated with subsequent laminitis appear to confirm that the risk for laminitis is positively correlated with the degree of hyperinsulinaemia, and is not conditional on a pathophysiological threshold, which calls for a cautious use of cut-offs in clinical cases.

Secondly, the study population is a major source of variability, which needs to be described on several levels. The breed and age are known factors affecting hyperinsulinaemia [43,44,45,46], as are previous feeding and the type of feeding [47,48,49], which is why feeding with grain or concentrate meals was avoided for several hours before testing in all but one study [37]. Furthermore, the proportion of horses affected by the outcome in the study population is critical. To provide an extreme example, a cut-off of 0 µIU/mL for insulin dysregulation will have an 80% accuracy and 100% sensitivity in a study population of 80% insulin-dysregulated horses, but no diagnostic value in a normal population, due to its null specificity.

Thirdly, the underlying statistical methods must be identified. The cut-off can be defined to maximize any performance metric, such as accuracy, sensitivity, etc. Youden’s index [50] is often used as a compromise between sensitivity and specificity, as the maximal value is reached when the maximum sum of sensitivity and specificity is attained. As explained above, this statistical consideration is directly linked to the proportion of individuals affected by the studied outcome in the study population. Nevertheless, lower (more sensitive) cut-offs for hyperinsulinaemia will generally be preferred to specific ones, as the balance of risks and costs is clearly in favour of laminitis prevention with exercise, nutritional management and, in select cases, medical treatment.

Despite all these differences, the published cut-offs for basal insulin values with laminitis as an outcome averaged at around 30 µIU/mL once converted to the Mercodia Equine assay [37,38,39]. The cut-offs targeting insulin resistance or ID were lower at around 10 to 20 µIU/mL (once converted) [34,35,36,41]. This is unsurprising, as the likelihood of laminitis incidence is correlated with the degree of hyperinsulinaemia, meaning that slightly insulin-resistant horses may not develop laminitis at all, even though they remain at risk.

While the app is useful for allowing such comparisons, cut-off values should not be applied blindly. In many cases, the authors of the above-cited works define an intermediate range for equivocal cases. Moreover, dynamic tests are much more reliable than basal testing in the diagnosis of ID and of the risk of laminitis development [6,51]. Clinical signs of obesity, previous laminitis, or PPID should be taken into account when interpreting test results in individual animals. Due to the variability in the horses’ response to testing, and the potential evolution of the disease, repeat testing is recommended in ambiguous cases.

## 5. Conclusions

We have presented an app (www.equine-insulin-converter.org, accessed on 2 July 2023) able to convert blood insulin measurements in horses from one assay to another, based on published and exclusive data on immunoassay comparisons. Comparisons between studies on the same assay pairs revealed non-negligible variation. This may affect the measurements obtained from different laboratories beyond the scope of our app, and should be acknowledged in the interpretation of diagnostic results. On the other hand, the reliability of our app is improved through the combination of multiple sources. Altogether, the app facilitates the detection of equids affected by ID and prone to laminitis, even when the assay used for establishing the cut-off is not available to the clinician. As demonstrated with cut-offs for basal blood insulin concentrations, it also enables meta-analyses of publications relying on different assays.

## Figures and Tables

**Figure 1 animals-13-02704-f001:**
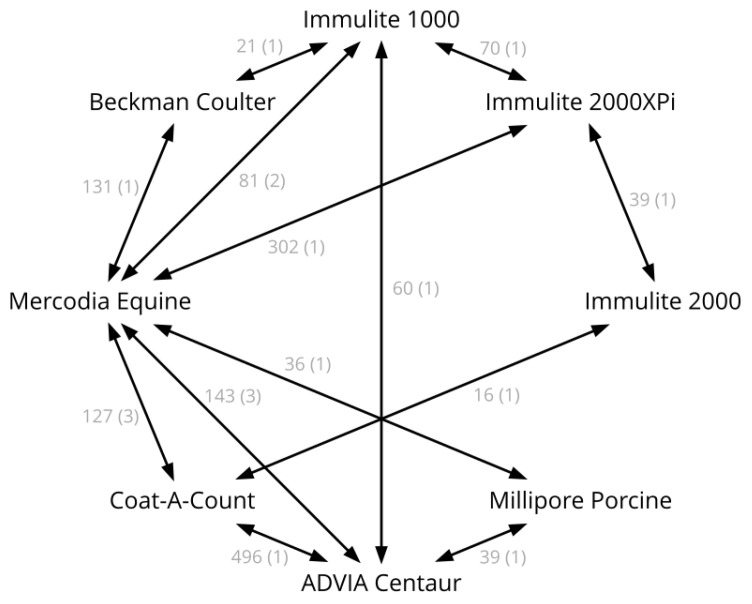
Diagram of the available assay comparisons. The number of samples is shown along the graph edges, followed by the number of studies in parentheses.

**Figure 2 animals-13-02704-f002:**
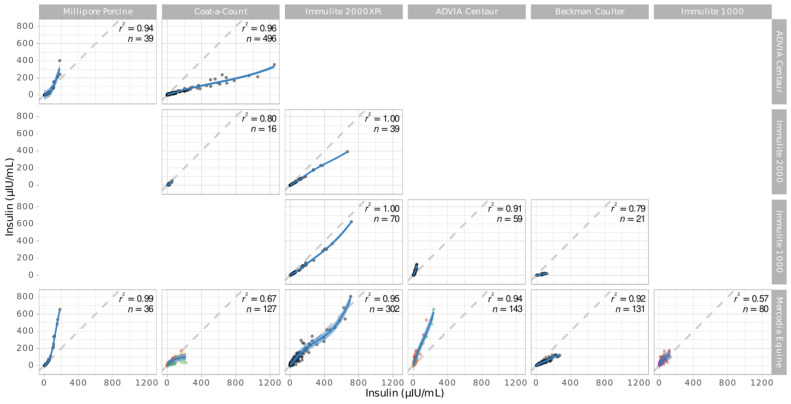
Scatterplot matrix of the collected assay comparison data, with the corresponding polynomials (dark blue) and their 95% prediction intervals (light blue). The *r*^2^ values obtained from the unweighted dataset and the sample sizes (n) are given alongside each polynomial. The dashed line in the background is the identity line. The assay names along the top and right side indicate which assays are being compared in the respective subplots. Different colours indicate that several data sources were used in the corresponding assay comparison.

**Figure 3 animals-13-02704-f003:**
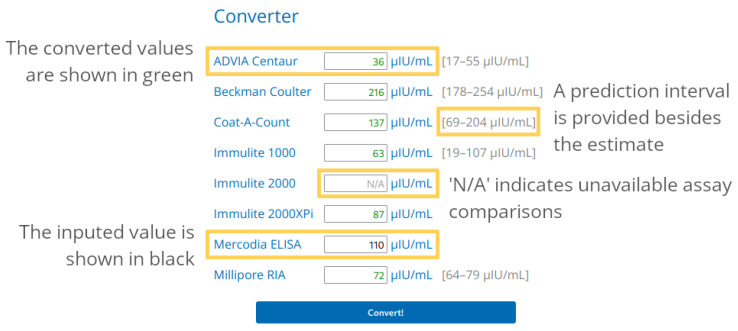
Annotated screenshot of the workhorse of the app, its “Converter” section. An insulin concentration can be entered in the field next to the desired assay. Once “Enter” or “Convert!” is hit, the estimates of the values that would have been obtained with other assays are calculated.

**Table 1 animals-13-02704-t001:** List of assays available on the web app.

	Assay Name	Assay Family
A	ADVIA Centaur Insulin Assay, Siemens Healthcare	CLIA
B	Coat-A-Count Insulin, Siemens Healthcare	RIA
C	Immulite 1000 Insulin Assay, Siemens Healthcare	CLIA
D	Immulite 2000 Insulin Assay, Siemens Healthcare	CLIA
E	Immulite 2000XPi Insulin Assay, Siemens Healthcare	CLIA
F	Insulin(e) IRMA KIT, Immunotech, Beckman Coulter	RIA
G	Equine Insulin ELISA, Mercodia	ELISA
H	Porcine Insulin RIA, Merck Millipore	RIA

**Table 2 animals-13-02704-t002:** List of data sources.

Source	Assays Compared	Number of Samples	Comments
Öberg et al., 2011 [15]	B vs. G	80	
Tinworth et al., 2011 [16]	B vs. G	18	
Borer-Weir et al., 2012 [17]	B vs. G	29	
Warnken et al., 2016 [19]	A vs. G A vs. H G vs. H	36 39 36	
Carslake et al., 2017 [20]	B vs. D	16	
Carslake et al., 2021 [26]	D vs. E	39	
De Laat et al., 2022 [34]	E vs. G	302	Raw data kindly provided by Boehringer Ingelheim
Durham, unpublished	C vs. E	70	Diagnostic samples
Fey, unpublished	A vs. B	496	Diagnostic samples
Warnken and Delarocque, unpublished	C vs. G A vs. G A vs. C	60 77 60	Ethics committee file number 33.8-42502-04-17/2646
Warnken, unpublished	C vs. G C vs. F F vs. G	21 21 131	Ethics committee file number 3.14-42502-04-13/1259
Warnken, unpublished	A vs. G	30	Diagnostic samples

**Table 3 animals-13-02704-t003:** Published cut-offs (COs) for basal insulin. All cut-offs were converted to the Mercodia Equine Insulin Assay (Conv. CO) using the app. The outcome for which, and the method through which, the cut-offs were determined, are presented (CO metric).

Reference	Assay	Feeding	Outcome	CO Metric	CO (µIU/mL)	Conv. CO (µIU/mL)
Lindase et al., 2021 [35]	Mercodia Equine	Fasted	EHC	Youden’s index	9.5	9.5
de Laat et al., 2022 [34]	Mercodia Equine	Fasted	Cluster	Balanced sens. and spec. ^1^	10.4	10.4
Olley et al., 2019 [36]	Immulite 1000 or 2000	Fasted	CGIT	Youden’s index	5.2	22
Menzies-Gow et al., 2017 [37]	Coat-a-Count	Unfasted	Laminitis	Youden’s index	21.8	26
Meier et al., 2018 [38]	ADVIA Centaur	Fasted	High NSC diet laminitis	CART	8.5	28
Carter et al., 2009 [39]	Coat-a-Count	Unfasted (hay)	Pasture associated laminitis	AUC in ROC	32	33
EEG 2020 [40]	Immulite 2000XPi	Unfasted (hay)			31	41
EEG 2020 [40]	Immulite 1000	Unfasted (hay)			20	49
Köller et al., 2016 [41]	Immulite 2000	Unfasted (hay)	Healthy	95% confidence interval ^2^	21	49

^1^ Minimal absolute difference between sensitivity and specificity. ^2^ Not a cut-off, but an upper reference range limit.

## Data Availability

All data used in the project are presented in the figures.

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
