# Peer review of "Development of a Web App to Convert Blood Insulin Concentrations among Various Immunoassays Used in Horses"

_animals, 2023, doi:10.3390/ani13172704_

Round 1

Reviewer 1 Report

Thank you for providing a well-researched and well-written manuscript as well as for developing a very practical tool for both equine researchers and practitioners alike. There are just two suggestions to consider.

Table 3. I assume the cut-offs are concentrations, so the CO and Conv CO columns require units.

Line 197. Should read 'these data were' neither presented or accounted for...

Quality of the English is fine

Author Response

Dear Reviewer,

Thank you for bringing these errors to our attention. We are glad that you are otherwise satisfied with our manuscript and thankful for your feedback. Both suggestions were implemented in the manuscript. Moreover, based on another reviewer's comments, we have added a sentence regarding back-conversions when using the app (l. 223). Please also note that the legend of Figure 2 was cut away in the initial version of the manuscript and was reinserted.

Reviewer 2 Report

Dear Authors, dear Editors,

The study "Development of a web app to convert blood insulin concentrations among various immunoassays used in horses" presents the development of an insulin converter whose algorithm was created on the basis of previously published and unpublished data (insulin concentrations) collected by the authors. The study is written clearly and understandably. The result of the study - the insulin converter- is a useful tool for veterinary medicine, both in the clinical field and in research. A strength of the study is the large number of samples thanks to a collaboration with different institutes and laboratories.

I have no suggestions for improvement with regard to the study design, the evaluation and presentation of the results and the discussion.

My only ambiguity is when using the app directly. Why do I get different results when, for example, I enter a value in a test and then "calculate back" the converted results? While the variances in the tests I've run don’t seem significant and are unlikely to be clinically relevant, I wonder if these inaccuracies might pose problems if the app is used for research purposes. Perhaps this should be discussed?

Author Response

Dear Reviewer,

Thank you very much for assessing our manuscript and app. Based on your suggestion a sentence regarding back conversions was added to the manuscript (l. 223). Please note that the legend of Figure 2 was cut away in the initial version of the manuscript and was reinserted. Otherwise only minor corrections were made.

Reviewer 3 Report

The study is interesting, and provides a necessary tool for equine clinics regarding insulin dysregulation. The text is well written, well organized, the introduction is short but provides the necessary information to understand the problem described. The material and method is well organized and well written. Likewise, the results of the study are well presented, as well as the figures and tables. On the other hand, the discussion is clear and orderly. In general, I have no objection to the manuscript being published in its present form.

Author Response

Dear Reviewer,

Thank you for your feedback. We are pleased that you are satified with our manuscript. Following changes were made after the first review round:

  • A sentence regarding back conversions was added to the manuscript (l. 223) based on another reviewer's comments
  • The legend of Figure 2 was cut away in the initial version of the manuscript and was reinserted
  • Minor corrections (missing units, orthographic corrections)